# Reinforcement-Enhanced Autoregressive Feature Transformation: Gradient-steered Search in Continuous Space for Postfix Expressions

**Dongjie Wang**[*]
Department of CS
University of Central Florida
dongjie.wang@ucf.edu

**Meng Xiao**[*]
CNIC, CAS
University of CAS
shaow@cnic.cn

**Min Wu**
Institute for Infocomm Research
A*STAR
wumin@i2r.a-star.edu.sg

**Pengfei Wang**
CNIC, CAS
University of CAS
wpf@cnic.cn

**Yuanchun Zhou**
CNIC, CAS
University of CAS
zyc@cnic.cn

**Yanjie Fu**[†]
School of Computing and AI
Arizona State University
yanjiefu@asu.edu

## Abstract

Feature transformation aims to generate new pattern-discriminative feature space from original features to improve downstream machine learning (ML) task performances. However, the discrete search space for the optimal feature explosively grows on the basis of combinations of features and operations from low-order forms to high-order forms. Existing methods, such as exhaustive search, expansion reduction, evolutionary algorithms, reinforcement learning, and iterative greedy, suffer from large search space. Overly emphasizing efficiency in algorithm design usually sacrifices stability or robustness. To fundamentally fill this gap, we reformulate discrete feature transformation as a continuous space optimization task and develop an embedding-optimization-reconstruction framework. This framework includes four steps: 1) reinforcement-enhanced data preparation, aiming to prepare high-quality transformation-accuracy training data; 2) feature transformation operation sequence embedding, intending to encapsulate the knowledge of prepared training data within a continuous space; 3) gradient-steered optimal embedding search, dedicating to uncover potentially superior embeddings within the learned space; 4) transformation operation sequence reconstruction, striving to reproduce the feature transformation solution to pinpoint the optimal feature space. Finally, extensive experiments and case studies are performed to demonstrate the effectiveness and robustness of the proposed method. The code and data are publicly accessible https://www.dropbox.com/sh/imh8ckui7va3k5u/AACulQegVx0MuywYyoCqSdVPa?dl=0.

## 1 Introduction

Feature transformation aims to derive a new feature space by mathematically transforming the original features to enhance the downstream ML task performances. However, feature transformation is usually manual, time-consuming, labor-intensive, and requires domain knowledge. These limitations motivate us to accomplish *Automated Feature Transformation* (**AFT**). AFT is a fundamental task because AFT can 1) reconstruct distance measures, 2) form a feature space with discriminative patterns, 3) ease machine learning, and 4) overcome complex and imperfect data representation.

There are two main challenges in solving AFT: 1) efficient feature transformation in a massive discrete search space; 2) robust feature transformation in an open learning environment. Firstly, it

---

[*]These authors have contributed equally to this work.
[†]Corresponding Author

37th Conference on Neural Information Processing Systems (NeurIPS 2023).

is computationally costly to reconstruct the optimal feature space from a given feature set. Such reconstruction necessitates a transformation sequence that contains multiple combinations of features and operations. Each combination indicates a newly generated feature. Given the extensive array of features and operations, the quantity of possible feature-operation combinations exponentially grows, further compounded by the vast number of potential transformation operation sequences. The efficiency challenge seeks to answer: *how can we efficiently identify the best feature transformation operation sequence?* Secondly, identifying the best transformation operation sequence is unstable and sensitive to many factors in an open environment. For example, if we formulate a transformation operation sequence as a searching problem, it is sensitive to starting points or the greedy strategy during iterative searching. If we identify the same task as a generation problem, it is sensitive to training data quality and the complexity of generation forms. The robustness challenge aims to answer: *how can we robustify the generation of feature transformation sequences?*

Prior literature partially addresses the two challenges. Existing AFT algorithms can be grouped into three categories: 1) expansion-reduction approaches [1, 2, 3], in which all mathematical operations are randomly applied to all features at once to generate candidate transformed features, followed by feature selection to choose valuable features. However, such methods are based on random generation, unstable, and not optimization-directed. 2) iterative-feedback approaches [4, 5, 6], in which feature generation and selection are integrated, and learning strategies for each are updated based on feedback in each iteration. Two example methods are Evolutionary Algorithms (EA) or Reinforcement Learning (RL) with downstream ML task accuracy as feedback. However, such methods are developed based on searching in a massive discrete space and are difficult to converge in comparison to solving a continuous optimization problem. 3) Neural Architecture Search (NAS)-based approaches [7, 8]. NAS was originally to identify neural network architectures using a discrete search space containing neural architecture parameters. Inspired by NAS, some studies formulated AFT as a NAS problem. However, NAS-based formulations are slow and limited in modeling all transformation forms. Existing studies show the inability to jointly address efficiency and robustness in feature transformation. Thus, we need a novel perspective to derive a novel formulation for AFT.

**Our Contribution: A Postfix Expression Embedding and Generation Perspective.** To fundamentally fill these gaps, we formulate the discrete AFT problem as a continuous optimization task and propose a reinforce**M**ent-enhanced aut**O**regressive fe**A**ture **T**ransformation framework, namely **MOAT**. To advance efficiency and robustness, this framework implements four steps: 1) reinforcement-enhanced training data preparation; 2) feature transformation operation sequence embedding; 3) gradient-steered optimal embedding search; 4) beam search-based transformation operation sequence reconstruction. Step 1 is to collect high-quality transformation operation sequence-accuracy pairs as training data. Specifically, we develop a cascading reinforcement learning structure to automatically explore transformation operation sequences and test generated feature spaces on a downstream predictor (e.g., decision tree). The self-optimizing policies enable agents to collect high-quality transformation operation sequences. The key insight is that when training data is difficult or cost expensive to collect, reinforcement intelligence can be used as an automated training data collector. Step 2 is to learn a continuous embedding space from transformation-accuracy training data. Specifically, we describe transformation operation sequences as postfix expressions, each of which is mapped into an embedding vector by jointly optimizing the transformation operation sequence reconstruction loss and accuracy estimation loss. Viewing feature transformation through the lens of postfix expressions offers the chance to mitigate the challenges associated with an exponentially expanding discrete search problem. By recasting the feature transformation operation sequence in postfix form, the search space becomes smaller, as each generation step involves selecting a single alphabetical letter pertaining to a feature or mathematical operation, rather than considering high-order expansions. Moreover, this postfix form captures feature-feature interaction information and empowers the generation model with the capacity to autonomously determine the optimal number and segmentation way of generated features. Step 3 is to leverage the gradient calculated from the improvement of the accuracy evaluator to guide the search for the optimal transformation operation sequence embedding. Step 4 is to develop a beam search-based generative module to reconstruct feature transformation operation sequences from embedding vectors. Each sequence is used to obtain a transformed feature space, and subsequently, the transformed space that yields the highest performance in the downstream ML task is identified as the optimal solution. Finally, we present extensive experiments and case studies to show the effectiveness and superiority of our framework.

## 2 Definitions and Problem Statement

### 2.1 Important Definitions

**Operation Set.** To refine the feature space for improving the downstream ML models, we need to apply mathematical operations to existing features to generate new informative features. All operations are collected in an operation set, denoted by $\mathcal{O}$. Based on the computation property, these operations can be classified as unary operations and binary operations. The unary operations such as "square", "exp", "log", etc. The binary operations such as "plus", "multiply", "minus", etc.

**Cascading Agent Structure.** We create a cascading agent structure comprised of *head feature agent*, *operation agent*, and *tail feature agent* to efficiently collect quantities of high-quality feature transformation records. The selection process of the three agents will share the state information and sequentially select candidate features and operations for refining the feature space.

**Feature Transformation Operation Sequence.** Assuming that $D = \{X, y\}$ is a dataset, which includes the original feature set $X = [f_1, \cdots, f_N]$ and predictive targets $y$. As shown in Figure 1, we transform the existing ones using mathematical compositions $\tau$ consisting of feature ID tokens and operations to generate new and informative features. $K$ transformation compositions are adopted to refine $X$ to a better feature space $\tilde{X} = [\tilde{f}_1, \cdots, \tilde{f}_K]$. The collection of the $K$ compositions refers to the feature transformation sequence, which is denoted by $\Gamma = [\tau_1, \cdots, \tau_K]$.

$$\tau_1 : (f_1 + (\frac{(f_1 + f_2)}{f_3}) - f_2), \tau_2 : (f_1 - (\sqrt{f_3})), \cdots, \tau_K : ((f_2)^2)$$

Figure 1: An example of feature transformation sequence: $\tau_{(.)}$ indicates the generated feature that is the combination of original features and mathematical operations.

### 2.2 Problem Statement

We aim to develop an effective and robust deep differentiable automated feature transformation framework. Formally, given a dataset $D = \{X, y\}$ and an operation set $\mathcal{O}$, we first build a cascading RL-agent structure to collect $n$ feature transformation accuracy pairs as training data, denoted by $R = \{(\Gamma_i, v_i)\}_{i=1}^{n}$, where $\Gamma_i$ is the transformation sequence and $v_i$ is the associated downstream predictive performance. We pursue two objectives thereafter: 1) building an optimal continuous embedding space for feature transformation sequences. We learn a mapping function $\phi$, a reconstructing function $\psi$, and an evaluation function $\omega$ to convert $R$ into a continuous embedding space $\mathcal{E}$ via joint optimization. In $\mathcal{E}$, each embedding point is associated with a feature transformation sequence and corresponding predictive performance. 2) identifying the optimal feature space. We adopt a gradient-based search to find the optimal feature transformation sequence $\Gamma^*$, given by:

$$\Gamma^* = \psi(\mathbf{E}^*) = argmax_{\mathbf{E} \in \mathcal{E}} \mathcal{A}(\mathcal{M}(\psi(\mathbf{E})(X)), y), \tag{1}$$

where $\psi$ can reconstruct a feature transformation sequence from any embedding point of $\mathcal{E}$; $\mathbf{E}$ is an embedding vector in $\mathcal{E}$ and $\mathbf{E}^*$ is the optimal one; $\mathcal{M}$ is the downstream ML model and $\mathcal{A}$ is the performance indicator. Finally, we apply $\Gamma^*$ to transform $X$ to the optimal feature space $X^*$ maximizing the value of $\mathcal{A}$.

## 3 Methodology

### 3.1 Framework Overview

Figure 2 shows the framework of MOAT including four steps: 1) reinforcement-enhanced transformation-accuracy data preparation; 2) deep postfix feature transformation operation sequence embedding; 3) gradient-ascent optimal embedding search; 4) transformation operation sequence reconstruction. In Step 1, a cascading agent structure consisting of two feature agents and one operation agent is developed to select candidate features and operators for feature crossing. The transformed feature sets are applied to a downstream ML task to collect the corresponding accuracy. The data collection process is automated and self-optimized by policies and feedback in reinforcement learning. We then convert these feature transformation operation sequences into postfix expressions. In Step 2, we develop an encoder-evaluator-decoder model to embed transformation operation sequence-accuracy pairs into a continuous embedding space by jointly optimizing the sequence reconstruction loss and performance evaluation loss. In detail, the encoder maps these transformation operation

sequences into continuous embedding vectors; the evaluator assesses these embeddings by predicting their corresponding model performance; the decoder reconstructs the transformation sequence using these embeddings. In Step 3, we first learn the embeddings of top-ranking transformation operation sequences by the well-trained encoder. With these embeddings as starting points, we search along the gradient induced by the evaluator to find the acceptable optimal embeddings with better model performances. In Step 4, the well-trained decoder then decodes these optimal embeddings to generate candidate feature transformation operation sequences through the beam search. We apply the feature transformation operation sequences to original features to reconstruct refined feature spaces and evaluate the corresponding performances of the downstream predictive ML task. Finally, the feature space with the highest performance is chosen as the optimal one.

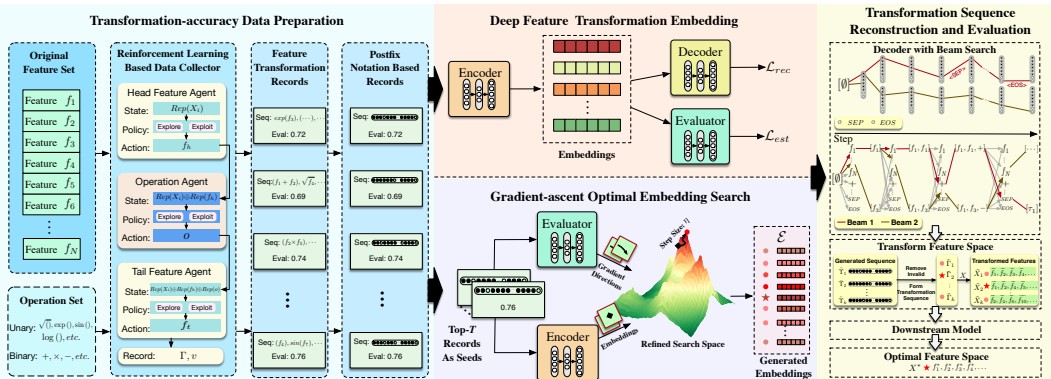

Figure 2: An overview of our framework. MOAT consists of four main components: 1) transformation-accuracy data preparation; 2) deep feature transformation embedding; 3) gradient-ascent optimal embedding search; 4) transformation sequence reconstruction and evaluation.

## 3.2    Reinforcement Training Data Preparation

**Why Using Reinforcement as Training Data Collector.** Our extensive experimental analysis shows that the quality of embedding space directly determines the success of feature transformation operation sequence construction. The quality of the embedding space is sensitive to the quality and scale of transformation sequence-accuracy training data: training data is large enough to represent the entire distribution; training data include high-performance feature transformation cases, along with certain random exploratory samples. Intuitively, we can use random sample features and operations to generate feature transformation sequences. This strategy is inefficient because it produces many invalid and low-quality samples. Or, we can use existing feature transformation methods (e.g., AutoFeat [2]) to generate corresponding records. However, these methods are not fully automated and produce a limited number of high-quality transformation records without exploration ability. We propose to view reinforcement learning as a training data collector to overcome these limitations.

**Reinforcement Transformation-Accuracy Training Data Collection.** Inspired by [4, 9], we formulate feature transformation as three interdependent Markov decision processes (MDPs). We develop a cascading agent structure to implement the three MDPs. The cascading agent structure consists of a head feature agent, an operation agent, and a tail feature agent. In each iteration, the three agents collaborate to select two candidate features and one operation to generate a new feature. Feedback-based policy learning is used to optimize the exploratory data collection to find diversified yet quality feature transformation samples. To simplify the description, we adopt the $i$-th iteration as an example to illustrate the reinforcement data collector. Given the former feature space as $X_i$, we generate new features $X_{i+1}$ using the head feature $f_h$, operation $o$, and tail feature $f_t$ selected by the cascading agent structure.

*1) Head feature agent.* This learning system includes: *State:* is the vectorized representation of $X_i$. Let $Rep(\cdot)$ be a state representation method, and the state can be denoted by $Rep(X_i)$. *Action:* is the head feature $f_h$ selected from $X_i$ by the reinforced agent.

*2) Operation agent.* This learning system includes: *State:* includes the representation of $X_i$ and the head feature, denoted by $Rep(X_i) \oplus Rep(f_h)$, where $\oplus$ indicates concatenation. *Action:* is the operation $o$ selected from the operation set $\mathcal{O}$.

*3) Tail feature agent.* This learning system includes: *State:* includes the representation of $X_i$, selected head feature, and operation, denoted by $Rep(X_i) \oplus Rep(f_h) \oplus Rep(o)$. *Action:* is the tail feature $f_t$ selected from $X_i$ by this agent.

*4) State representation method, $Rep(\cdot)$.* For the representation of the feature set, we employ a descriptive statistical technique to obtain the state with a fixed length. In detail, we first compute the descriptive statistics (*i.e.* count, standard deviation, minimum, maximum, first, second, and third quantile) of the feature set column-wise. Then, we calculate the same descriptive statistics on the output of the previous step. After that, we can obtain the descriptive matrix with shape $\mathbb{R}^{7 \times 7}$ and flatten it as the state representation with shape $\mathbb{R}^{1 \times 49}$. For the representation of the operation, we adopt its one-hot encoding as $Rep(o)$.

*5) Reward function.* To improve the quality of the feature space, we use the improvement of a downstream ML task performance as the reward. Thus, it can be defined as: $\mathcal{R}(X_i, X_{i+1}) = \mathcal{A}(\mathcal{M}(X_{i+1}), y) - \mathcal{A}(\mathcal{M}(X_i), y)$.

*6) Learning to Collect Training Data.* To optimize the entire procedure, we minimize the mean squared error of the Bellman Equation to get a better feature space. During the exploration process, we can collect amounts of high-quality records $(\Gamma, v)$ for constructing an effective continuous embedding space, where $\Gamma$ is the transformation sequence, and $v$ is the downstream model performance.

### 3.3 Postfix Expressions of Feature Transformation Operation Sequences

**Why Transformation Operation Sequences as Postfix Expressions.** After training data collection, a question arises: how can we organize and represent these transformation operation sequences in a computationally-tangible and machine-learnable format?

Figure 3 (a) shows an example of a feature transformation operation sequence with two generated features. To convert this sequence into a machine-readable expression, a naive idea is to enclose each calculation in a pair of brackets to indicate its priority (Figure 3(b)). But, the representation of Figure 3(b) has four limitations: (1) *Redundancy.* Many priority-related brackets are included to ensure the unambiguous property and correctness of mathematical calculations. (2) *Semantic Sparsity.* The quantity of bracket tokens can dilute the semantic information of the sequence, making model convergence diffi-

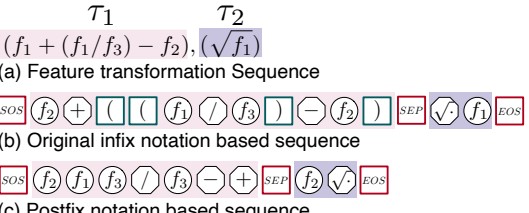

(a) Feature transformation Sequence

(b) Original infix notation based sequence

(c) Postfix notation based sequence

Figure 3: Compared to infix-based expressions, postfix-based expressions optimize token redundancy, enhance semantics, prevent illegal transformations, and minimize the search space.

cult. (3) *Illegal Transformation.* If the decoder makes one mistake on bracket generation, the entire generated sequence will be wrong. (4) *Large Search Space.* The number of combinations of features and operations from low-order to high-order interactions is large, making the search space too vast.

**Using Postfix Expression to Construct Robust, Concise, and Unambiguous Sequences.** To address the aforementioned limitations, we convert the transformation operation sequence $\Gamma$ into a postfix-based sequence expression. Specifically, we scan each mathematical composition $\tau$ in $\Gamma$ from left to right and convert it from the infix-based format to the postfix-based one. We then concatenate each postfix expression by the <SEP> token and add the <SOS> and <EOS> tokens to the beginning and end of the entire sequence. Figure 3(c) shows an example of such a postfix sequence. We denote it as $\Upsilon = [\gamma_1, \cdots, \gamma_M]$, where each element is a feature ID token, operation token, or three other unique tokens. The detailed pseudo code of this conversion process is provided in Appendix A.

The postfix sequences don't require numerous brackets to ensure the calculation priority. We only need to scan each element in the sequence from left to right to reconstruct corresponding transformed features. Such a concise and short expression can reduce sequential modeling difficulties and computational costs. Besides, a postfix sequence indicates a unique transformation process, thus, reducing the ambiguity of the feature transformation sequence. Moreover, the most crucial aspect is the reduction of the search space from exponentially growing discrete combinations to a limited token set $\mathcal{C}$ that consists of the original feature ID tokens, operation tokens, and other three unique tokens. The length of the token set is $|\mathcal{O}| + |X| + 3$, where $|\mathcal{O}|$ is the number of the operation set, $|X|$ is the dimension of the original feature set, and 3 refers to the unique tokens <SOS>, <SEP>, <EOS>.

**Data Augmentation for Postfix Transformation Sequences.** Big and diversified transformation sequence records can benefit the learning of a pattern discriminative embedding space. Be sure to notice that, a feature transformation sequence consists of many independent segmentations, each of which is the composition of feature ID and operation tokens and can be used to generate a new feature. These independent segmentations are order-agnostic. Our idea is to leverage this property to conduct data augmentation to increase the data volume and diversity. For example, given a transformation operation sequence and corresponding accuracy $\{\Upsilon, v\}$, we first divide the postfix expression into different segmentations by <SEP>. We then randomly shuffle these segmentations and use <SEP> to concatenate them together to generate new postfix transformation sequences. After that, we pair the new sequences with the corresponding model accuracy performance to improve data diversity and data volume for better model training and to create the continuous embedding space.

## 3.4 Deep Feature Transformation Embedding

After collecting and converting large-scale feature transformation training data to a set of postfix expression-accuracy pairs $\{(\Upsilon_i, v_i)\}_{i=1}^n$, we develop an encoder-evaluator-decoder structure to map the sequential information of these records into an embedding space. Each embedding vector is associated with a transformation operation sequence and its corresponding model accuracy.

**Encoder $\phi$:** The Encoder aims to map any given postfix expression to an embedding (a.k.a., hidden state) $\mathbf{E}$. We adopt a single layer long short-term memory [10] (LSTM) as Encoder and acquire the continuous representation of $\Upsilon$, denoted by $\mathbf{E} = \phi(\Upsilon) \in \mathbf{R}^{M \times d}$, where $M$ is the total length of input sequence $\Upsilon$ and $d$ is the hidden size of the embedding.

**Decoder $\psi$:** The Decoder aims to reconstruct the postfix expression of the feature transformation operation sequence $\Upsilon$ from the hidden state $\mathbf{E}$. In MOAT, we set the backbone of the Decoder as a single-layer LSTM. For the first step, $\psi$ will take an initial state (denoted as $h_0$) as input. Specifically, in step-$i$, we can obtain the decoder hidden state $h_i^d$ from the LSTM. We use the dot product attention to aggregate the encoder hidden state and obtain the combined encoder hidden state $h_i^e$. Then, the distribution in step-$j$ can be defined as: $P_\psi(\gamma_i|\mathbf{E}, \Upsilon_{<i}) = \frac{\exp(W_{\gamma_i}(h_i^d \oplus h_i^e))}{\sum_{c \in \mathcal{C}} \exp(W_c(h_i^d \oplus h_i^e))}$, where $\gamma_i \in \Upsilon$ is the $i$-th token in sequence $\Upsilon$, and $\mathcal{C}$ is the token set. $W$ stand for the parameter of the feedforward network. $\Upsilon_{<i}$ represents the prediction of the previous or initial step. By multiplying the probability in each step, we can form the distribution of each token in $\Upsilon$, given as: $P_\psi(\Upsilon|\mathbf{E}) = \prod_{i=1}^M P_\psi(\gamma_i|\mathbf{E}, \Upsilon_{<i})$. To make the generated sequence similar to the real one, we minimize the negative log-likelihood of the distribution, defined as: $\mathcal{L}_{rec} = -\log P_\psi(\Upsilon|\mathbf{E})$.

**Evaluator $\omega$:** The Evaluator is designed to estimate the quality of continuous embeddings. Specifically, we will first conduct mean pooling on $\mathbf{E}$ by column to aggregate the information and obtain the embedding $\bar{\mathbf{E}} \in \mathbf{R}^d$. Then $\bar{\mathbf{E}}$ is input into a feedforward network to estimate the corresponding model performance, given as: $\hat{v} = \omega(\mathbf{E})$. To minimize the gap between estimated accuracy and real-world gold accuracy, we leverage the Mean Squared Error (MSE) given by: $\mathcal{L}_{est} = MSE(v, \omega(\mathbf{E}))$.

**Joint Training Loss $\mathcal{L}$:** We jointly optimize the encoder, decoder, and evaluator. The joint training loss can be formulated as: $\mathcal{L} = \alpha\mathcal{L}_{rec} + (1-\alpha)\mathcal{L}_{est}$, where $\alpha$ is the trade-off hyperparameter that controls the contribution of sequence reconstruction and accuracy estimation loss.

## 3.5 Gradient-Ascent Optimal Embedding Search

To conduct the optimal embedding search, we first select top-$T$ transformation sequences ranked by the downstream predictive accuracy. The well-trained encoder is then used to embed these postfix expressions into continuous embeddings, which later will be used as seeds (starting points) of gradient ascent. Assuming that one search seed embedding is $\mathbf{E}$, we search, starting from $\mathbf{E}$, toward the gradient direction induced by the evaluator $\omega$: $\tilde{\mathbf{E}} = \mathbf{E} + \eta\frac{\partial \omega}{\partial \mathbf{E}}$, where $\tilde{\mathbf{E}}$ denotes the refined embedding, $\eta$ is the size of each searching step. The model performance of $\tilde{\mathbf{E}}$ is supposed to be better than $\mathbf{E}$ due to $\omega(\tilde{\mathbf{E}}) \geq \omega(\mathbf{E})$. For $T$ seeds, we can obtain the enhanced embeddings $[\tilde{\mathbf{E}}_1, \tilde{\mathbf{E}}_2, \cdots, \tilde{\mathbf{E}}_T]$.

## 3.6 Transformation Operation Sequence Reconstruction and Evaluation

We reconstruct the transformation sequences by the well-trained decoder $\psi$ using the collected candidate (i.e., acceptable optimal) embeddings $[\tilde{\mathbf{E}}_1, \tilde{\mathbf{E}}_2, \cdots, \tilde{\mathbf{E}}_T]$. This process can be denoted by: $[\tilde{\mathbf{E}}_1, \tilde{\mathbf{E}}_2, \cdots, \tilde{\mathbf{E}}_T] \xrightarrow{\psi} \{\tilde{\Upsilon}_i\}_{i=1}^T$. To identify the best transformation sequence, we adopt the

beam search strategy [11, 12, 13] to generate feature transformation operation sequence candidates. Specifically, given a refined embedding $\tilde{\mathbf{E}}$, at step-t, we maintain the historical predictions with beam size $b$, denoted as $\{\Upsilon_{<t}^i\}_{i=1}^b$. For the $i$-th beam, the probability distribution of the token identified by the well-trained decoder $\psi$ at the $t$-th step is is $\gamma$, which can be calculated as follows: $P_t^i(\gamma) = P_\psi(\gamma|\tilde{\mathbf{E}}, \tilde{\Upsilon}_{<t}^i) * P_\psi(\tilde{\Upsilon}_{<t}^i|\tilde{\mathbf{E}})$, where the probability distribution $P_t^i(\gamma)$ is the continued multiplication between the probability distribution of the previous decoding sequence and that of the current decoding step. We can collect the conditional probability distribution of all tokens for each beam. After that, we append tokens with top-$b$ probability values to the historical prediction of each beam to get a new historical set $\{\tilde{\Upsilon}_{<t+1}^i\}_{i=1}^b$. We can iteratively conduct this decoding process until confronted with the <EOS> token. We select the transformation sequence with the highest probability value as output. Hence, $T$ enhanced embeddings may produce $T$ transformation sequences $\{\tilde{\Upsilon}_i\}_{i=1}^T$. We divide each of them into different parts according to the <SEP> token and check the validity of each part and remove invalid ones. Here, the validity measures whether the mathematical compositions represented by the postfix part can be successfully calculated to produce a new feature. These valid postfix parts reconstruct a feature transformation operation sequence $\{\tilde{\Gamma}_i\}_{i=1}^T$, which are used to generate refined feature space $\{\tilde{X}_i\}_{i=1}^T$. Finally, we select the feature set with the highest downstream ML performance as the optimal feature space $\mathbf{X}^*$.

## 4   Experiments

This section reports the results of both quantitative and qualitative experiments that were performed to assess MOAT with other baseline models. All experiments were conducted on AMD EPYC 7742 CPU, and 8 NVIDIA A100 GPUs. For more platform information, please refer to Appendix B.2.

### 4.1   Datasets and Evaluation Metrics

We used 23 publicly available datasets from UCI [14], LibSVM [15], Kaggle [16], and OpenML [17] to conduct experiments. The 23 datasets involve 14 classification tasks and 9 regression tasks. Table 1 shows the statistics of these datasets. We used F1-score, Precision, Recall, and ROC/AUC to evaluate classification tasks. We used 1-Relative Absolute Error (1-RAE) [4], 1-Mean Average Error (1-MAE), 1-Mean Square Error (1-MSE), and 1-Root Mean Square Error (1-RMSE) to evaluate regression tasks. We used the Valid Rate to evaluate the transformation sequence generation. A valid sequence means it can successfully conduct mathematical compositions without any ambiguity and errors. The valid rate is the average of all correct sequence numbers divided by the total number of generated sequences. The greater the valid rate is, the superior the model performance is. Because it indicates that the model can capture the complex patterns of mathematical compositions and search for more effective feature transformation sequences.

### 4.2   Baseline Models

We compared our method with eight widely-used feature generation methods: (1) **RDG** generates feature-operation-feature transformation records at random for generating new feature space; (2) **ERG** first applies operation on each feature to expand the feature space, then selects the crucial features as new features. (3) **LDA** [18] is a matrix factorization-based method to obtain the factorized hidden state as the generated feature space. (4) **AFAT** [2] is an enhanced version of ERG that repeatedly generate new features and use multi-step feature selection to select informative ones. (5) **NFS** [7] models the transformation sequence of each feature and uses RL to optimize the entire feature generation process. (6) **TTG** [5] formulates the transformation process as a graph, then implements an RL-based search method to find the best feature set. (7) **GRFG** [4] uses three collaborated reinforced agents to conduct feature generation and proposes a feature grouping strategy to accelerate agent learning. (8) **DIFER** [8] embeds randomly generated feature transformation records with a seq2seq model, then employs gradient search to find the best feature set. MOAT and DIFER belong to the same setting. We demonstrate the differences between them in Appendix **??**. Besides, we developed two variants of MOAT in order to validate the impact of each technical component: (i) **MOAT**$^{-d}$ replaces the RL-based data collection component with collecting feature transformation-accuracy pairs at random. (ii) **MOAT**$^{-a}$ removes the data augmentation component. We randomly split each dataset into two independent sets. The prior 80% is used to build the continuous embedding space and the remaining 20% is employed to test transformation performance. This experimental setting avoids any test data leakage and ensures a fairer transformation performance comparison. We adopted Random Forest as the downstream machine learning model. Because it is a robust, stable, well-tested method, thus,

Table 1: Overall performance comparison. 'C' for binary classification, and 'R' for regression. The best results are highlighted in **bold**. The second-best results are highlighted in underline. (**Higher values indicate better performance.**)

| Dataset | Source | C/R | Samples | Features | RDG | ERG | LDA | AFAT | NFS | TTG | GRFG | DIFER | MOAT |
|---------|--------|-----|---------|----------|-----|-----|-----|------|-----|-----|------|-------|------|
| Higgs Boson | UCIrvine | C | 50000 | 28 | 0.695 | 0.702 | 0.513 | 0.697 | 0.691 | 0.699 | 0.707 | 0.669 | **0.712** |
| Amazon Employee | Kaggle | C | 32769 | 9 | 0.932 | 0.934 | 0.916 | 0.930 | 0.932 | 0.933 | 0.932 | 0.929 | **0.936** |
| PimaIndian | UCIrvine | C | 768 | 8 | 0.760 | 0.761 | 0.638 | 0.765 | 0.749 | 0.745 | 0.754 | 0.760 | **0.807** |
| SpectF | UCIrvine | C | 267 | 44 | 0.760 | 0.757 | 0.665 | 0.760 | 0.792 | 0.760 | 0.818 | 0.766 | **0.912** |
| SVMGuide3 | LibSVM | C | 1243 | 21 | 0.787 | 0.826 | 0.652 | 0.795 | 0.792 | 0.798 | 0.812 | 0.773 | **0.849** |
| German Credit | UCIrvine | C | 1001 | 24 | 0.680 | 0.740 | 0.639 | 0.683 | 0.687 | 0.645 | 0.683 | 0.656 | **0.730** |
| Credit Default | UCIrvine | C | 30000 | 25 | 0.805 | 0.803 | 0.743 | 0.804 | 0.801 | 0.798 | 0.806 | 0.796 | **0.810** |
| Messidor_features | UCIrvine | C | 1150 | 19 | 0.624 | 0.669 | 0.475 | 0.665 | 0.638 | 0.655 | 0.692 | 0.660 | **0.749** |
| Wine Quality Red | UCIrvine | C | 999 | 12 | 0.466 | 0.461 | 0.433 | 0.480 | 0.462 | 0.467 | 0.470 | 0.476 | **0.559** |
| Wine Quality White | UCIrvine | C | 4900 | 12 | 0.524 | 0.510 | 0.449 | 0.516 | 0.525 | 0.531 | 0.534 | 0.507 | **0.536** |
| SpamBase | UCIrvine | C | 4601 | 57 | 0.906 | 0.917 | 0.889 | 0.912 | 0.925 | 0.919 | 0.922 | 0.912 | **0.932** |
| AP-omentum-ovary | OpenML | C | 275 | 10936 | 0.832 | 0.814 | 0.658 | 0.830 | 0.832 | 0.758 | 0.849 | 0.833 | **0.885** |
| Lymphography | UCIrvine | C | 148 | 18 | 0.108 | 0.144 | 0.167 | 0.150 | 0.152 | 0.148 | 0.182 | 0.150 | **0.267** |
| Ionosphere | UCIrvine | C | 351 | 34 | 0.912 | 0.921 | 0.654 | 0.928 | 0.913 | 0.902 | 0.933 | 0.905 | **0.985** |
| Housing Boston | UCIrvine | R | 506 | 13 | 0.404 | 0.409 | 0.020 | 0.416 | 0.425 | 0.396 | 0.404 | 0.381 | **0.467** |
| Airfoil | UCIrvine | R | 1503 | 5 | 0.519 | 0.519 | 0.220 | 0.521 | 0.519 | 0.500 | 0.521 | 0.558 | **0.629** |
| Openml_618 | OpenML | R | 1000 | 50 | 0.472 | 0.561 | 0.052 | 0.472 | 0.473 | 0.467 | 0.562 | 0.408 | **0.692** |
| Openml_589 | OpenML | R | 1000 | 25 | 0.509 | 0.610 | 0.011 | 0.508 | 0.505 | 0.503 | 0.627 | 0.463 | **0.656** |
| Openml_616 | OpenML | R | 500 | 50 | 0.070 | 0.193 | 0.024 | 0.149 | 0.167 | 0.156 | 0.372 | 0.076 | **0.526** |
| Openml_607 | OpenML | R | 1000 | 50 | 0.521 | 0.555 | 0.107 | 0.516 | 0.519 | 0.522 | 0.621 | 0.476 | **0.673** |
| Openml_620 | OpenML | R | 1000 | 25 | 0.511 | 0.546 | 0.029 | 0.527 | 0.513 | 0.512 | 0.619 | 0.442 | **0.642** |
| Openml_637 | OpenML | R | 500 | 50 | 0.136 | 0.152 | 0.043 | 0.176 | 0.152 | 0.144 | 0.307 | 0.072 | **0.465** |
| Openml_586 | OpenML | R | 1000 | 25 | 0.568 | 0.624 | 0.110 | 0.543 | 0.544 | 0.544 | 0.646 | 0.482 | **0.700** |

\* We reported F1-Score for classification tasks, and 1-RAE for regression tasks.

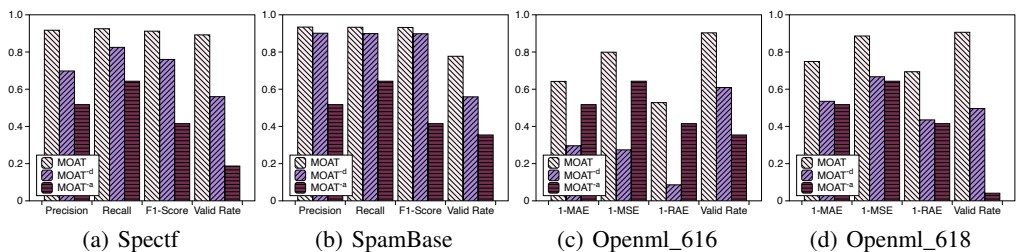

(a) Spectf  (b) SpamBase  (c) Openml_616  (d) Openml_618

Figure 4: The influence of data collection (MOAT$^{-d}$) and data augmentation (MOAT$^{-a}$) in MOAT.

we can reduce performance variation caused by the model, and make it easy to study the impact of feature space. We provided more experimental and hyperparameter settings in Appendix B.1.

### 4.3 Performance Evaluation

**Overall Performance** This experiment aims to answer: *Can* MOAT *effectively generate transformation sequence for discovering optimal feature space with excellent performance?* Table 1 shows the overall comparison between MOAT and other models in terms of F1-score and 1-RAE. We noticed that MOAT beats others on all datasets. The underlying driver is that MOAT builds an effective embedding space to preserve the knowledge of feature transformation, making sure the gradient-ascent search module can identify the best-transformed feature space following the gradient direction. Another interesting observation is that MOAT significantly outperforms DIFER and has a more stable performance. There are two possible reasons: 1) The high-quality transformation records produced by the RL-based data collector provide a robust and powerful foundation for building a discriminative embedding space; 2) Postfix notation-based transformation sequence greatly decreases the search space, making MOAT easily capture feature transformation knowledge. Thus, this experiment validates the effectiveness of MOAT.

**Data collection and augmentation.** This experiment aims to answer: *Is it essential to conduct data collection and augmentation to maintain the performance of* MOAT*?* To achieve this goal, we developed two model variants of MOAT: 1) MOAT$^{-d}$, which randomly collects feature transformation records for continuous space construction; 2) MOAT$^{-a}$, which removes the data augmentation step in MOAT. We select two classification tasks (i.e., Spectf and SpamBase) and two regression tasks (i.e., Openml_616 and Openml_618) to show the comparison results. The results are reported in the Figure 4. Firstly, we found that the performance of MOAT is much better than MOAT$^{-d}$. The underlying driver is that high-quality transformation records collected by the RL-based collector build a robust foundation for embedding space learning. It enables gradient-ascent search to effectively identify the optimal feature space. Moreover, we observed that the performance of MOAT$^{-a}$ is inferior to MOAT. This observation reflects that limited data volume and diversity cause the construction

Table 3: Time complexity comparison between MOAT and DIFER

| Dataset | Model Name | Data Collection 512 instances | Model Training | Solution Searching | Performance |
|---|---|---|---|---|---|
| Wine Red | MOAT | 921.6 | 5779.2 | 101.2 | 0.559 |
| Wine White | MOAT | 2764.8 | 7079.6 | 103.4 | 0.536 |
| Openml_618 | MOAT | 4556.8 | 30786.1 | 111.3 | 0.692 |
| Openml_589 | MOAT | 4044.8 | 8942.3 | 105.2 | 0.656 |
| Wine Red | DIFER | 323.2 | 11 | 180 | 0.476 |
| Wine White | DIFER | 1315.8 | 33 | 624 | 0.507 |
| Openml_618 | DIFER | 942.3 | 33 | 534 | 0.408 |
| Openml_589 | DIFER | 732.5 | 157 | 535 | 0.463 |

of embedding space to be unstable and noisy. Thus, this experiment shows the necessity of the data collection and augmentation components in MOAT.

**Beam search.** This experiment aims to answer *What is the influence of beam search for improving the valid rate of generated transformation sequences?* To observe the impacts of beam search, we set the beam size as 5 and 1 respectively, and add DIFER as another comparison object. We compare the generation performance in terms of valid rate. Figure 5 shows the comparison results in terms of valid rate. We noticed that the 5-beams search outperforms the 1-beam search. The underlying driver is that the

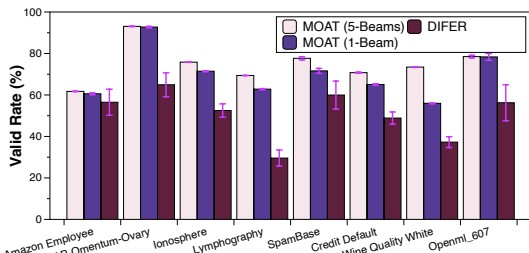

Figure 5: The influence of beam size in MOAT.

increased beam size can identify more legal and reasonable transformation sequences. Another interesting observation is that DIFER is significantly worse than MOAT and its error bar is longer. The underlying driver is that DIFER collects transformation records at random and it generates transformation sequences using greedy search. The collection and generation ways involve more random noises, distorting the learned embedding space.

**Robustness check.** This experiment aims to answer: *Is* MOAT *robust to different downstream machine learning models?* We replaced the downstream ML models with Random Forest (RF), XGBoost (XGB), Support Vector Machine (SVM), K-Nearest Neighborhood (KNN), Ridge, LASSO, and Decision Tree (DT) to observe the variance of model performance respectively. Table 2 shows the comparison results on Spectf in terms of the F1-score. We observed that MOAT keeps the best performance regard-

Table 2: Robustness check of MOAT with distinct ML models on Spectf dataset in terms of F1-score.

| | RF | XGB | SVM | KNN | Ridge | LASSO | DT |
|---|---|---|---|---|---|---|---|
| RDG | 0.760 | 0.818 | 0.750 | 0.792 | 0.718 | 0.749 | 0.864 |
| ERG | 0.757 | 0.813 | 0.753 | 0.766 | 0.778 | 0.750 | 0.790 |
| LDA | 0.665 | 0.715 | 0.760 | 0.749 | 0.759 | 0.760 | 0.665 |
| AFAT | 0.760 | 0.808 | 0.722 | 0.759 | 0.723 | 0.770 | 0.844 |
| NFS | 0.792 | 0.799 | 0.732 | 0.792 | 0.744 | 0.749 | 0.864 |
| TTG | 0.760 | 0.819 | 0.765 | 0.750 | 0.716 | 0.749 | 0.842 |
| GRFG | 0.818 | 0.842 | 0.580 | 0.760 | 0.729 | 0.744 | 0.786 |
| DIFER | 0.766 | 0.794 | 0.727 | 0.777 | 0.647 | 0.744 | 0.809 |
| MOAT | **0.912** | **0.897** | **0.876** | **0.916** | **0.780** | **0.844** | **0.929** |

less of downstream ML models. A possible reason is that the RL-based data collector can customize the transformation records based on the downstream ML model. Then, the learned embedding space may comprehend the preference and properties of the ML model, thereby resulting in a globally optimal feature space. Thus, this experiment shows the robustness of MOAT.

**Time complexity analysis.** This experiment aims to answer *What is the time complexity of each technical component of* MOAT? To accomplish this, we selected the SOTA model DIFER as a comparison baseline. Figure 3 shows the time costs comparison of data collection, model training, and solution searching in terms of seconds. We let both MOAT and DIFER collect 512 instances in the data collection phase. We found that the increased time costs for MOAT occur in the data collection and model training phases. However, once the model converges, MOAT 's inference time is significantly reduced. The underlying driver is that the RL-based collector spends more time gathering high-quality data, and the sequence formulation for the entire feature space increases the learning time cost for the sequential model. But, during inference, MOAT outputs the entire feature transformation at once, whereas DIFER requires multiple reconstruction iterations based on the generated feature space's dimension. The less inference time makes MOAT more practical and suitable for real-world scenarios.

To thoroughly analyze the multiple characteristics of MOAT, we also analyzed the space complexity (See Appendix C.1), model scalability (See Appendix C.2), and parameter sensitivity (See Appendix C.3). Meanwhile, we provided two qualitative analyses: learned embedding visualization (See Appendix C.4) and traceability case study (See Appendix C.5).

## 5   Related Works

**Automated Feature Transformation (AFT)** can enhance the feature space by automatically transforming the original features via mathematical operations [19, 20]. Existing works can be divided into three categories: 1) expansion-reduction based approaches [1, 3, 21, 2, 22]. Those methods first expand the original feature space by explicitly [23] or greedily [24] decided mathematical transformation, then reduce the space by selecting useful features. However, they are hard to produce or evaluate both complicated and effective mathematical compositions, leading to inferior performance. 2) evolution-evaluation approaches [4, 5, 6, 25, 9, 26]. These methods integrate feature generation and selection into a closed-loop learning system. They iteratively generate effective features and keep the significant ones until they achieve the maximum iteration number. The entire process is optimized by evolutionary algorithms or reinforcement learning models. However, they still focus on how to simulate the discrete decision-making process in feature engineering. Thus, they are still time-consuming and unstable. 3) Auto ML-based approaches [7, 8]. Auto ML aims to find the most suitable model architecture automatically [27, 28, 29, 30]. The success of auto ML in many area [31, 32, 33, 34, 35, 36] and the similarity between auto ML and AFT inspire researchers to formulate AFT as an auto ML task to resolve. However, they are limited by: 1) incapable of producing high-order feature transformation; and 2) unstable transformation performance. Recent studies tried to convert the feature transformation problem into a continuous space optimization task to search for the optimal feature space efficiently. *DIFER* [8] is a cutting-edge method that is comparable to our work in problem formulation. However, the following constraints limit DIFER's practicality: 1) DIFER collects transformation-accuracy data at random, resulting in many invalid training data with inconsistent transformation performances; 2) DIFER embeds and reconstructs each transformed feature separately and, thus, ignores feature-feature interactions; 3) DIFER needs to manually decide the number of generated features, making the reconstruction process more complicated. 4) the greedy search for transformation reconstruction in DIFER leads to suboptimal transformation results. To fill these gaps, we first implement an RL-based data collector to automate high-quality transformation record collection. We then leverage the postfix expression idea to represent the entire transformation operation sequence to model feature interactions and automatically identify the number of reconstructed features. Moreover, we employ beam search to advance the robustness, quality, and validity of transformation operation sequence reconstruction.

## 6   Conclusion Remarks

In this paper, we propose an automated feature transformation framework, namely MOAT. In detail, we first develop an RL-based data collector to gather high-quality transformation-accuracy pairs. Then, we offer an efficient postfix-based sequence expression way to represent the transformation sequence in each pair. Moreover, we map them into a continuous embedding space using an encoder-decoder-evaluator model structure. Finally, we employ a gradient-ascent search to identify better embeddings and then use beam search to reconstruct the transformation sequence and identify the optimal one. Extensive experiments show that the continuous optimization setting can efficiently search for the optimal feature space. The RL-based data collector is essential to keep an excellent and stable transformation performance. The postfix expression sequence enables MOAT to automatically determine the transformation depth and length, resulting in more flexible transformation ways. The beam search technique can increase the validity of feature transformation. The most noteworthy research finding is that the success of MOAT indicates that the knowledge of feature transformation can be embedded into a continuous embedding space to search for better feature space. Thus, it inspires us to regard the learning paradigm as the backbone to develop a large feature transformation model and quickly fine-tune it for different sub-tasks, which is also our future research direction.

## 7   Acknowledgments

This research was partially supported by the National Science Foundation (NSF) via grant numbers: 2040950, 2006889, and 2045567.

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

# Appendix

## A   Postfix Transformation Conversion

Algorithm 1 is the pseudo-code for the convert the original feature transformation sequence $\Gamma$ to the postfix notation based sequence $\Upsilon$. In detail, we first initialize a list $\Upsilon$ and two stacks $S_1$ and $S_2$, respectively. For each element $\tau$ in $\Gamma$, we scan it from left to right. When getting a feature ID token, we push it to $S_2$. When receiving a left parenthesis, we push it to $S_1$. When obtaining any operations, we pop each element in $S_1$ and push them into $S_2$ until the last component of $S_1$ is the left bracket. Then we push this operation into $S_1$ When getting a right parenthesis, we pop every element from $S_1$ and then push into $S_2$ until we confront a left bracket. Then we remove this left bracket from the top of $S_1$. When the end of the input $\tau$ encounters, we append every token from $S_2$ into $\Upsilon$. If this $\tau$ is not the last element in $\Gamma$, we will append a <SEP> token to indicate the end of this $\tau$. After we process every element in $\Gamma$, we add <SOS> and <EOS> tokens to the beginning and end of $\Upsilon$ to form the postfix notation-based transformation sequence $\Upsilon = [\gamma_1, \cdots, \gamma_M]$. Each element in $\Upsilon$ is a feature ID token, operation token, or three other special tokens. We convert each transformation sequence through this algorithm and construct the training set with them.

---

**Algorithm 1:** Postfix transformation sequence conversion

  **input** :Feature transformation sequence $\Gamma$
  **output**:Postfix notation based transformation sequence $\Upsilon$

1 $\Upsilon \longleftarrow \emptyset$;
2 **for** $\tau \leftarrow \Gamma$ **do**
3   $S_1, S_2 \longleftarrow \emptyset, \emptyset$;
4   **for** $\gamma \leftarrow \tau$ **do**
5     **if** $\gamma$ **is** left bracket **then**
6       $S_1.push(\gamma)$;
7     **else if** $\gamma$ **is** right bracket **then**
8       **while** $t \leftarrow S_1.pop()$ **is not** left bracket **do**
9         $S_2.push(t)$;
10     **else if** $\gamma$ **is** operation **then**
11       **while** $S_1.peek()$ **is not** left bracket **do**
12         $S_2.push(S_1.pop())$;
13       $S_1.push(\gamma)$;
14     **else**
15       $S_2.push(\gamma)$;
16   **while** $S_2$**not** $= \emptyset$ **do**
17     $\Upsilon.append(S_2.pop(0))$;
18   **if** $\tau$ **is not** *the last element* **then**
19     $\Upsilon.append($<SEP>$)$;
20 Add <SOS> and <EOS> to the head and tail of $\Upsilon$ respectively;
21 **return** $\Upsilon$

---

## B   Experimental Settings and Reproducibility

### B.1   Hyperparameter Settings and Reproducibility

The operation set consists of *square root, square, cosine, sine, tangent, exp, cube, log, reciprocal, quantile transformer, min-max scale, sigmoid, plus, subtract, multiply, divide*. For the data collection part, we ran the RL-based data collector for 512 epochs to collect a large amount of feature transformation-accuracy pairs. For the data augmentation part, we randomly shuffled each transformation sequence 12 times to increase data diversity and volume. We adopted a single-layer LSTM

Table 4: Space complexity comparison on MOAT with different dataset

| Dataset | Sample Number | Column Number | Parameter Size |
|---|---|---|---|
| Airfoil | 1503 | 5 | 139969 |
| Amazon employee | 32769 | 9 | 138232 |
| ap_omentum_ovary | 275 | 10936 | 155795 |
| german_credit | 1001 | 24 | 141127 |
| higgs | 50000 | 28 | 141899 |
| housing boston | 506 | 13 | 139004 |
| ionosphere | 351 | 34 | 143057 |
| lymphography | 148 | 18 | 139969 |
| messidor_features | 1150 | 19 | 140162 |
| openml_620 | 1000 | 25 | 141320 |
| pima_indian | 768 | 8 | 138039 |
| spambase | 4601 | 57 | 147496 |
| spectf | 267 | 44 | 144987 |
| svmguide3 | 1243 | 21 | 140548 |
| uci_credit_card | 30000 | 25 | 141127 |
| wine_red | 999 | 12 | 138618 |
| wine_white | 4900 | 12 | 138618 |
| openml_586 | 1000 | 25 | 141320 |
| openml_589 | 1000 | 25 | 141320 |
| openml_607 | 1000 | 50 | 146145 |
| openml_616 | 500 | 50 | 146145 |
| openml_618 | 1000 | 50 | 146145 |
| openml_637 | 500 | 50 | 146145 |

as the encoder and decoder backbones and utilized 3-layer feed-forward networks to implement the predictor. The hidden state sizes of the encoder, decoder and predictor are 64, 64, and 200, respectively. The embedding size of each feature ID token and operation token was set to 32. To train MOAT, we set the batch size as 1024, the learning rate as 0.001, and $\lambda$ as 0.95 respectively. For inferring new transformation sequences, we used top-20 records as the seeds with beam size 5.

## B.2 Experimental Platform Information

All experiments were conducted on the Ubuntu 18.04.6 LTS operating system, AMD EPYC 7742 CPU, and 8 NVIDIA A100 GPUs, with the framework of Python 3.9.10 and PyTorch 1.8.1.

## C Experimental Results

### C.1 Space Complexity Analysis.

To analyze the space complexity of MOAT, we illustrate the parameter size of MOAT when confronted with different datasets. Table 4 shows the comparison results. We can find that the model size of MOAT keeps relatively stable without significant fluctuations. The underlying driver is that the encoder-evaluator-decoder learning paradigm can embed the knowledge of discrete sequences with variant lengths into a fixed-length embedding vector. Thus, such an embedding process can make the parameter size to be stable instead of increasing as the data size grows. Thus, this experiment indicates that MOAT has good scalability when confronted with different scaled datasets.

### C.2 Scalability Check

We visualized the changing trend of the time cost of searching for better feature spaces over sample size and feature dimensions of different datasets. Figure 6 shows the comparison results. We found that the time cost of MOAT keeps stable with the increase of sample size of the feature set. A possible reason is that MOAT only focuses on the decision-making benefits of feature ID and operation tokens instead of the information of the entire feature set, making the searching process sample size irrelevant. Another interesting observation is that the search time is still stable although the feature dimension of the feature set varies significantly. A possible explanation is that we map transformation records of varying lengths into a continuous space with a constant length. The searching time in this space is input dimensionality-agnostic. Thus, this experiment shows the MOAT has excellent scalability.

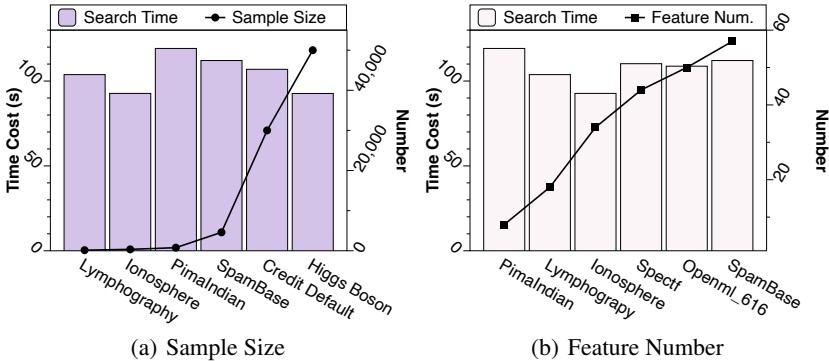

(a) Sample Size        (b) Feature Number

Figure 6: Scalability check of MOAT in search time based on sample size and feature number.

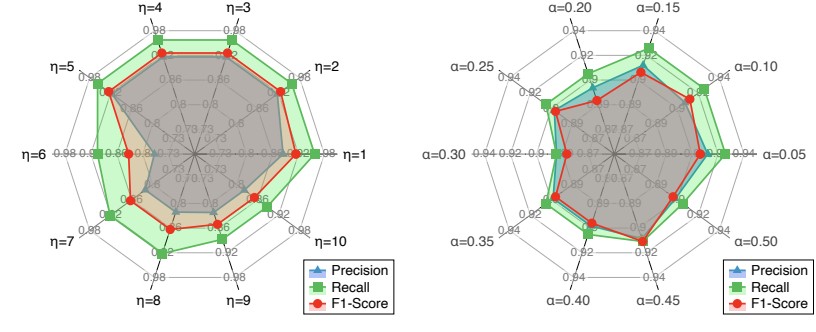

(a) Search Step Size $\eta$ (under $\alpha = 0.05$)     (b) Training Trade-off $\alpha$ (under $\eta = 1$)

Figure 7: Parameter sensitivity on search step size $\eta$ and trade-off parameter $\alpha$ on Spectf dataset.

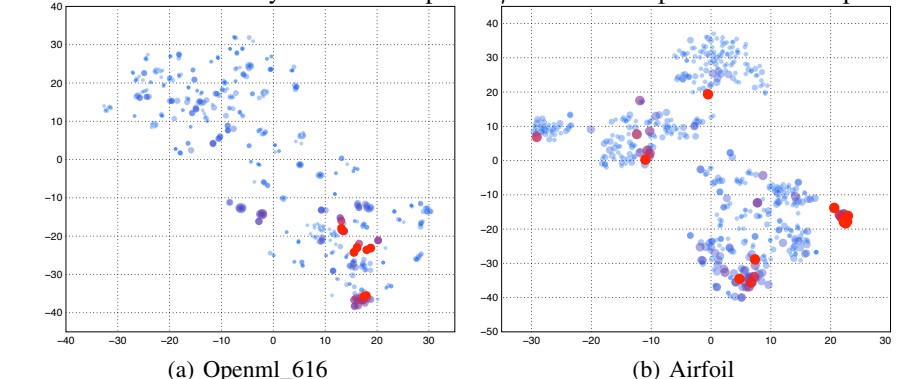

(a) Openml_616          (b) Airfoil

Figure 8: The visualization of learned transformation sequence embedding (from MOAT).

### C.3 Parameter sensitivity analysis

To validate the parameter sensitivity of the search step size $\eta$ (See section 3.5) and the trade-off parameter $\alpha$ in the training loss (See section 3.4), we set the value of $\eta$ from 1 to 10, and set the value of $\alpha$ from 0.05 to 0.50 to observe the difference. Figure 7 shows the comparison results in terms of precision, recall, and F1-score. When the search step size grows, the downstream ML performance initially improves, then declines marginally. A possible reason for this observation is that a too-large step size may make the gradient-ascent search algorithm greatly vibrate in the continuous space, leading to missing the optimal embedding point and transformed feature space. Another interesting observation is that the standard deviation of the model performance is lower than $0.01$ under different parameter settings. This observation indicates that MOAT is not sensitive to distinct parameter settings. Thus, the learning and searching process of MOAT is robust and stable.

### C.4 Learned embedding Analysis.

We selected Airfoil and Openml_616 as examples to visualize their learned continuous embedding space. In detail, we first collected the latent embeddings generated by the transformation records. Then, we use T-SNE to map them into a 2-dimensional space for visualization. Figure 8 shows

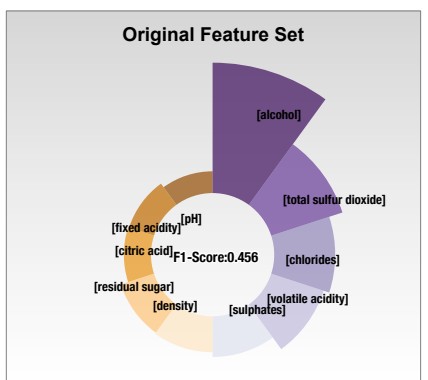
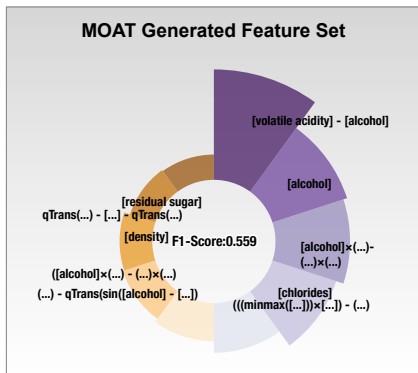

(a) Original Feature Space  (b) MOAT Generated Feature Space

Figure 9: Comparison of traceability on the original feature space and the MOATgenerated one.

the visualization results, in which each point represents a unique feature transformation sequence. The size of each point means its downstream performance. The bigger point size indicates that the downstream performance is superior. We colored the top 20 embedding points according to the performance in red. We found that the distribution locations of the top 20 embedding points are different. A potential reason is that the corresponding transformation sequences of the top 20 embedding points are different lengths. The sequence reconstruction loss distributes them to different areas of the embedding space. Moreover, we observed that the top 20 embedding points are close in the space even though the positions are different. The underlying driver is that the estimation loss makes these points with good performance clustered. Thus, this case study reflects that the reconstruction loss and estimation loss make the continuous space associate the transformation sequence and the corresponding model performance.

## C.5   Traceability case study

We selected the top 10 essential features for prediction in the original, and MOAT transformed feature space of the Wine Quality Red dataset for comparison. Figure 9 shows the comparison results. The texts associated with each pie chart are the corresponding feature name. The larger the pie area is, the more critical the feature is. We found that almost 70% critical features in the new feature space are generated by MOAT and they improve the downstream ML performance by 22.6%. This observation indicates that MOAT really comprehends the properties of the feature set and ML models in order to produce a more effective feature space. Another interesting finding is that '[alcohol]' is the essential feature in the original feature set. But MOAT generates more mathematically composited features using '[alcohol]'. This observation reflects that MOAT not only can capture the significant features but also produce more effective knowledge for enhancing the model performance. Such composited features can make domain experts trace their ancestor resources and summarize new analysis rules for evaluating the quality of red wine.

