# OpenReview forum: "Reinforcement-Enhanced Autoregressive Feature Transformation: Gradient-steered Search in Continuous Space for Postfix Expressions"
_NeurIPS.cc/2023/Conference — NeurIPS 2023 spotlight_

### Official Review · Reviewer_GfrD · 2023-07-03

**Soundness:** 3 good
**Presentation:** 4 excellent
**Contribution:** 3 good
**Rating:** 6
**Confidence:** 4

**Summary:**

Feature transformation is an effective way to improve downstream task. This paper aims to improve the search efficiency of the optimal feature space while ensuring the stability and robustness of the transformation. They formulate the discrete Automated Feature Transformation (AFT) problem as a continuous optimization task and propose a reinforcement-enhanced autoregressive feature Transformation framework (MOAT). MOAT implements four steps to advance efficiency and robustness. Extensive experiments and case studies are performed to demonstrate the effectiveness and robustness of the proposed method.

**Strengths:**

This paper provides a clear and detailed illustration of the proposed framework, especially the modules in the framework. They conduct extensive experiments to demonstrate the superiority of the framework, including ablation studies for each module, to demonstrate the efficiency and robustness of the framework.

**Weaknesses:**

Missing reference to experimental results on page 8, Data collection and augmentation section.
Several methods have used reinforcement learning to improve search efficiency at present (e.g., in data augmentation, NAS), and the framework proposed in this paper lacks novelty to some extent.

**Questions:**

(1) MOAT uses multi-stage process to obtain a convincing performance. Is it possible to learn an optimal feature space using reinforcement learning by end-to-end?
(2) Does the optimal feature space searching problem can be directly solved by representation learning in continuous space, rather than combining multiple operations in discrete space？

**Limitations:**

This paper pays more attention to framework design and lacks some theoretical perspectives. In addition, how to design the framework proposed in this paper into an end-to-end training process is desirable.

---

> ### Author Rebuttal · Authors · 2023-08-07
>
> Dear Reviewer GfrD
>
> Thank you for your detailed feedback and observations on our work. We highly appreciate the recognition of the strengths of our paper, and we acknowledge the points raised in the weaknesses and questions sections. We will address each of them comprehensively:
>
> **1. Regarding the End-to-end Model Architecture:**
>
> Response: The end-to-end reinforcement learning process for optimal feature space learning has been explored in works such as GRFG, NFS, and other reinforced methodologies. However, our extensive research and experimentation showed that while such an approach promises to streamline, it poses notable challenges, particularly regarding training stability and convergence. To illustrate, consider GRFG, which, despite its architecture mirroring the RL-based data collector, demands over 3000 epochs to converge, often settling for sub-optimal results. In stark contrast, our MOAT framework swiftly learns a continuous search space from merely RL-based data collectors' initial 512 search records. This generates a more stable, effective, and efficient feature transformation sequence.
>
>
> **2. Regarding the Merit of Representation Learning:**
>
> Response: Representation learning is a cornerstone in modern machine learning, offering a transformative approach to feature extraction and data interpretation. At its core, representation learning provides an avenue to automatically discover the representations needed for data analysis tasks, bypassing manual feature engineering, which has historically been time-consuming and domain-specific.
> In our study, we search for another way to extract the feature information from the dataset, which is the core of Data-centric AI, bringing automation into feature engineering.
> Our approach could easily apply to some small datasets, and this small amount of data might be insufficient to train a deep neural network for representation learning. In contrast, our approach can generate meaningful features by conducting mathematical feature-feature crossing. The fully traceable generated feature also can be summarized as domain knowledge by its operated mathematical transformation.
>
> **3. Regarding Novelty or Contribution**
>
> Response: Our computing insights and technical contributions go beyond the combination of GRFG and DIFER. The following implications and findings are beneficial for future research:
>
> We highlight a postfix expression embedding and generation perspective for feature transformation.
>
> We demonstrate that reinforcement intelligence can be used as a self-optimizing training data collector to explore high-quality feature transformations, evaluate downstream ML task accuracy, and automate training data collection.
>
> We find that integrating transformation sequence reconstruction loss and downstream task accuracy estimation loss can better measure the effectiveness of an embedding space and strengthen the denoising capability of gradient-based search of the optimal transformation embedding.
>
> We encode a feature transformation operation sequence as a single postfix expression so that the generative model can capture feature-feature interactions and automatically identify the dimensionality of the new feature space and the complexity of the transformation operation sequence.
>
> Our framework automatically determines the dimension of the generated feature space by examining the special token <EOS>, contrasting with DIFER's manual setting requirement.
>
> These insights are proven to improve the reliability and performance of automated feature engineering tasks.
>
> **4. Regarding the Missing Reference on Page 8:**
>
> Response: We apologize for this oversight. We will revise this issue in the final version.

---

### Official Review · Reviewer_yQs6 · 2023-07-05

**Soundness:** 4 excellent
**Presentation:** 3 good
**Contribution:** 4 excellent
**Rating:** 7
**Confidence:** 5

**Summary:**

This paper introduces a succinct yet effective framework for automatic feature transformation. The authors mapped decision sequences collected from reinforcement learning into a high-dimensional vector space. Optimization is carried out through the gradient direction provided by an evaluator, and a sequence reconstruction component is employed to rebuild the decision sequence of this feature transformation. Overall, the paper's experiments are comprehensive, and the discussions provide excellent validation of various features of the framework. The comparisons with existing models also aptly place this work among the latest contributions in the field.

**Strengths:**

1. The design of the entire framework is clear and comprehensible, and based on the authors' descriptions and experimental discussions, this framework also shows potential for reuse in other fields.

2. The experimental design is detailed and particularly addresses my concerns about the time-consuming nature of using reinforcement learning for data collection.

3. Similar to the second strength described above, a significant issue with using reinforcement learning in feature transformation tasks is that it might require an excessive number of search steps to achieve optimal. However, this framework can use a fixed number of search steps to construct the space for searching and achieve respectable downstream task performance, significantly mitigating the problem of uncertain duration in feature transformation tasks based on reinforcement learning.

**Weaknesses:**

1. There are a few typos in this paper, e.g., the legend of the figure 4, the model name is wrong.

2. The authors have proposed an encoder-decoder model based on LSTM for sequence modeling. However, there is no discussion on why this specific method was chosen. It would be beneficial if the authors could explain the rationale behind this choice. Could other sequence modeling methods, such as Transformers, have been considered or used instead?

3. While the design of the model and ample experiments mutually validate the feasibility of this framework, if there could be more in-depth insights provided, it might help establish this work as foundational for broader fields.

**Questions:**

1. The framework's description of the data collection part for the optimization objective lacks clarity. The authors should consider adding more detail in the appendix to better elucidate this aspect.

2. Are there any way to make the framework more generalized, e.g., extend this method to graph-like data, picuture-like data.

3. Would it be feasible to design a more generalized operation set that views these mathematical operations as analogous to sub-networks?

---

> ### Author Rebuttal · Authors · 2023-08-07
>
> Dear Reviewer yQs6,
>
> We sincerely appreciate the reviewer's time and effort in assessing our paper. We are pleased to find that the design and potential of the framework were recognized as strengths. We want to address the highlighted weaknesses and answer the posed questions to improve our paper's quality and clarity.
>
> **1. Regarding Typos and Minor Revision:**
>
> Response: We apologize for the oversight and will promptly correct the typos and the error in the legend of Figure 4 in our next revision.
>
> **2. Regarding the Choice of LSTM:**
>
> Response: Our framework is a generic feature learning framework, and the sequential model's purpose is to preserve feature learning knowledge in a continuous space, providing a solid foundation for subsequent steps. LSTM, as a vanilla model structure, effectively demonstrates the framework's generalization ability. However, if desired, our framework can also accommodate other sequential models, e.g., Transformers.
>
> **3. Regarding the Optimization Objective of the Data Collection Component:**
>
> Response: We realize that the data collection part may require more clarity. We will provide a more detailed description in the main paper and, if space-constrained, in the appendix.
>
> **4. Regarding the Framework Generalization on graph-like, picture-like data:**
>
> Response: It's an interesting point. Currently, our method is designed with specific data in mind, i.e., tabular data. However, extending it to handle graph-like or picture-like data is possible, and we are in the initial stages of exploring these extensions.
>
> **5. Regarding a Generalized Operation Set:**
>
> Response: Designing a set of operations that treat mathematical operations as analogous to sub-networks is a compelling idea. While our current framework has not incorporated this, we see its merit and will consider it for future work or iterations of the framework.
>
> We once again thank the reviewer for their constructive comments and will ensure to make the necessary revisions to enhance the paper's quality and impact.

---

### Official Review · Reviewer_miDK · 2023-07-06

**Soundness:** 3 good
**Presentation:** 3 good
**Contribution:** 4 excellent
**Rating:** 7
**Confidence:** 4

**Summary:**

The authors propose a feature transformation method that reformulates the problem as a continuous space optimization task and utilizes a reinforcement-enhanced autoregressive framework for gradient-steered search. The method involves four steps: (1) reinforcement-enhanced data preparation, (2) feature transformation operation sequence embedding, (3) gradient-steered optimal embedding search, and (4) transformation operation sequence reconstruction. The proposed method is evaluated through extensive experiments and case studies, demonstrating its effectiveness and robustness.

**Strengths:**

1. This paper proposes a novel automated feature transformation framework that has not been explored before, and it achieves significant improvement in performance compared to previous methods.

2. This paper provides clear and detailed explanations of the proposed method's four-step process, making it easy for readers to follow and replicate the method.

3. This paper conducts extensive experiments and case studies to illustrate the effectiveness of the framework from different perspectives.

4. The authors released the related code and data, which can help other researchers reproduce the experiments.

5. This paper provides clear and detailed explanations of the proposed method's four-step process, making it easy for readers to follow and replicate the method.

**Weaknesses:**

1. The authors use LSTM as the backbone of their framework. Any reason for choosing LSTM? How about other alternatives, such as Transformer?

2. There are some typos in this paper. For instance, in Figure 4 (c) and Figure 4 (d), the name of the model variant should be MOAT^-a instead of GBFG^-a. The authors should fix these typos for keeping consistency.

3. Analyzing the experimental results, the RL-based data collector is important. But, the description for this part is limited. Can the authors provide more explanation on this?


**Questions:**

1. It is unclear why the performance of DIFER is not good for openml 616, openml 637. Can you provide some explanation on this?

2. After reviewing the appendix section, I found that MOAT requires more training time but less inference time than DIFER. Can the authors explain the reason for it?

---

> ### Author Rebuttal · Authors · 2023-08-07
>
> Dear Reviewer miDK,
>
> We want to thank you for acknowledging this paper's novelty, effectiveness, extensive experiments, and reproducibility.
>
> **1. Regarding the inferior performance of DIFER on the openML dataset:**
>
> Response: OpenML616 and OpenML637 are two artificially constructed datasets, some of which contain added noise columns. In practice, MOAT will obtain high-quality samples through a reinforcement learning-based feature transformation sequence generator and use these samples as seeds to produce better feature transformation sequences. As the reinforcement learning agent converges, these artificially added noises can be easily removed. In contrast, DIFER selects better features from a set of generated random sequences for its search. This could potentially retain noise sequences, affecting downstream tasks' performance.
>
> **2. Regarding the difference between training and inference time for MOAT and DIFER**
>
> Response: This observation is due to the two frameworks' strategies to search for optimal feature transformation sequences. In DIFER, it models and searches \textbf{each column} as an independent transformation sequence. This means the sequence length it needs to model is much shorter than MOAT. Still, its search requires generation for each column according to a given sequence length hyperparameter, resulting in more search operations. MOAT models and searches the transformation operation sequence for the entire dataset, where the model determines the lengths of different columns, so the time consumption for sequence modeling is higher. Still, the time consumption for the generation process is less.
>
>
> **3. Regarding the description of RL-based data collector:**
>
> Response: We realized that we had omitted too much in the section describing the optimization objectives of RL-based data collectors, which might lead to misunderstandings. In the final version, we will detail the overall optimization objectives of reinforcement learning.
>
> **4. Some minor revisions:**
>
> Response: In the future version, we will address the typo and other minor errors in Figure 4.

---

### Official Review · Reviewer_MJDJ · 2023-07-07

**Soundness:** 4 excellent
**Presentation:** 3 good
**Contribution:** 4 excellent
**Rating:** 7
**Confidence:** 5

**Summary:**

Distinct from existing work, this study collects feature transformation operation sequences, which have a well-researched background based on reinforcement learning. It then obtains hidden representations of these limited numbers of sequences in a self-supervised manner and finally optimizes this continuous vector representation to generate superior sequences guided by gradients. Specifically, to model the sequences, the authors propose an LSTM-based encoder-decoder-evaluator architecture with just a few collected samples. Detailed runtime analysis was conducted in this work to demonstrate its advantages over reinforcement learning-based (GRFG) and random generation-based (DIFER) approaches. The experiments, coupled with the authors' analytical discussions, effectively substantiate the viewpoints declared in the paper.

**Strengths:**

1) This paper is well-written and easy to understand, with an appropriate level of detail in describing the methods. It considers the primary issue, time consumption, associated with this framework in terms of experimental design, effectively addressing the readers' concerns.

2) The paper's model architecture is thoughtfully designed, and the authors articulate their motivations for each component with great clarity. The streamlined nature of the entire framework demonstrates the depth of the authors' understanding and thoughtful consideration of this work.

3) The authors provide thorough experiments and analysis for this work, covering aspects of interest in this field such as performance, robustness, runtime, scalability, memory usage, traceability, and efficacy tests for each component. The visualization of the hidden space well illustrates the reasons for the model's effectiveness in conducting gradient search.

**Weaknesses:**

1) In the sections 2.1 Important Definitions and Section 3.2 Reinforcement Training Data Preparation, the authors have inconsistencies in the naming of the agents within the cascading agent structure for the data collection part. For instance, it's named as 'feature agent1' in the definition section, while it's referred to as 'head feature agent' in the methodology part. The authors should maintain consistency in these definition names to reduce confusion for the readers.

2) The description of the optimization objective for reinforcement learning in the data collection section is not detailed enough. The authors should provide a more comprehensive description of this critical component.


**Questions:**

1) Why does DIFER require more time in the inference phase? The authors should provide a detailed explanation for this.

2) In the time complexity analysis, the authors have chosen four datasets - Wine Red, Wine White, Openml_618, and OpenML_589. However, they have not adequately described the rationale behind selecting these particular datasets.

3) There are some avoidable writing errors, such as on page 13 of the appendix where the authors incorrectly referred to Table 3 as Figure 3. The authors should conduct a thorough check of the entire paper to prevent inconsistencies in naming.

---

> ### Author Rebuttal · Authors · 2023-08-07
>
> Dear Reviewer MJDJ,
>
> Thank you for acknowledging that our paper has a solid research background, well-written, easy-to-understand, and extensive experiments to support the claimed research insights and technical contributions. For your mentioned issues:
>
> **1. Regarding the typos, name inconsistency, and figure caption errors:**
>
> Response: We will revise the correlated description in Section 2.2 Problem Statement to avoid the inconsistency of the name definitions. We will thoroughly check before the final version and prevent hidden typos and errors.
>
> **2. Regarding the details of the optimization objective for the reinforcement learning component:**
>
> Response: We tried to present a balanced description of the model architecture and provide an as light as possible data collection component description, but based on your suggestion, we realized that we had omitted too much in the section describing the optimization objectives, which might lead to misunderstandings. In the final version, we will detail the overall optimization objectives of reinforcement learning.
>
> **3. Regarding the time complexity analysis of DIFER:**
>
> Response: DIFER implements a completely different sequence generation strategy. After the model converges, DIFER will generate each column in a fixed number of steps and then use a filter to remove invalid sequences generated. In contrast, MOAT will directly generate the entire sequence, where the model determines the lengths of different columns, and the model will try to generate valid sequences through Beam Search. Such differences lead to MOAT having better Inference Time than DIFER when generating datasets with the same number of columns.
>
> **4. Regarding the time complexity analysis of MOAT:**
>
> Response: We neglected to explain why the datasets were selected in the main text. According to Table 1, Wine Red and Wine White have the same number of features but different numbers of samples. Comparing the time efficiency of MOAT on these two datasets can reflect the model's scalability on datasets with different instances. Conversely, Openml618 and Openml589 have the same number of instances but different numbers of features, and comparing the time efficiency of these datasets can reflect the model's scalability on datasets with varying feature counts. These two experiments combined can effectively validate the model's scalability. We realize that the lack of this part of the description may affect the reader's understanding, and we will detail the reasons for these dataset selections in the final version.

---

### Decision · Program_Chairs · 2023-09-21

**Decision:**

Accept (spotlight)

**Comment:**

This paper studies the problem of feature transformation, which is very important for downstream tasks. This paper formulates the discrete automated feature transformation as a continuous optimization problem and proposes a novel transformation framework. Extensive experiments are provided, which can show the efficiency, and robustness of this framework. Moreover, the code and datasets are provided.